# Homemade Nucleic Acid Preservation Buffer Proves Effective in Preserving the Equine Faecal Microbiota over Time at Ambient Temperatures

**DOI:** 10.3390/ani13193107

**Published:** 2023-10-05

**Authors:** Ashley B. Ward, Patricia A. Harris, Caroline McG. Argo, Christine Watson, Madalina Neacsu, Wendy R. Russell, Antonio Ribeiro, Elaina Collie-Duguid, Zeynab Heidari, Philippa K. Morrison

**Affiliations:** 1School of Veterinary Medicine, Scotland’s Rural College, Aberdeen AB21 9YA, UK; 2The Rowett Institute, University of Aberdeen, Foresterhill, Aberdeen AB25 2ZD, UK; 3School of Medicine Medical Sciences and Nutrition, University of Aberdeen, Foresterhill, Aberdeen AB25 2ZD, UK; 4Equine Studies Group, Waltham Petcare Science Institute, Leicestershire LE14 4RT, UK; 5Department of Rural Land Use, Scotland’s Rural College, Aberdeen AB21 9YA, UK; 6Centre for Genome-Enabled Biology and Medicine, University of Aberdeen, King’s College, Aberdeen AB24 3FX, UK

**Keywords:** microbiota, equine faeces, sample storage, DNA preservation, 16S rRNA, gene survey

## Abstract

**Simple Summary:**

The microbial community in horse faeces can be assessed to make inferences about the gut bacteria, which is linked to the animals’ health. However, faecal bacterial communities can shift over time if not preserved between the points of sampling and processing, which could cause misleading results. This study stored equine faecal samples under four preservation treatments at room temperature for up to 150 h and assessed the resulting impact on the samples’ bacterial communities. Treatments included “COLD” (samples packaged with a cool pack), “CLX” (2% chlorhexidine digluconate solution), “NAP” (nucleic acid preservation buffer), and “FTA” (Whatman FTA™ cards). Samples were assessed after storage for 0, 24, 72, and 150 h at room temperature under the different treatments. The results showed that NAP buffer was effective in preserving the most prominent features of the equine faecal bacterial community for up to 150 h at room temperature, but the processing of FTA cards was inadequate to capture the full bacterial profile present. The cold preservation, CLX, and NAP treatments were equally effective in maintaining the bacterial community in equine faecal samples for up to 24 h. These findings demonstrate the effectiveness of NAP buffer and the potential of using COLD and CLX treatments for sample preservation at room temperature. This study also showed changes in the bacteria found in equine faeces that occur under preservation for up to 150 h.

**Abstract:**

The equine faecal microbiota is often assessed as a proxy of the microbial community in the distal colon, where the microbiome has been linked to states of health and disease in the horse. However, the microbial community structure may change over time if samples are not adequately preserved. This study stored equine faecal samples from *n* = 10 horses in four preservation treatments at room temperature for up to 150 h and assessed the resulting impact on microbial diversity and the differential abundance of taxa. Treatments included “COLD” (samples packaged with a cool pack), “CLX” (2% chlorhexidine digluconate solution), “NAP” (nucleic acid preservation buffer), and “FTA” (Whatman FTA™ cards). The samples were assessed using 16S rRNA gene sequencing after storage for 0, 24, 72, and 150 h at room temperature under the different treatments. The results showed effective preservation of diversity and community structure with NAP buffer but lower diversity (*p* = 0.001) and the under-representation of Fibrobacterota in the FTA card samples. The NAP treatment inhibited the overgrowth of bloom taxa that occurred by 72 h at room temperature. The COLD, CLX, and NAP treatments were effective in preserving the faecal microbiota for up to 24 h at room temperature, and the CLX and NAP treatments improved the yield of Patescibacteria and Fibrobacterota in some cases. The cold and CLX treatments were ineffective in preventing community shifts that occurred by 72 h at room temperature. These findings demonstrate the suitability of the COLD, NAP, and CLX treatments for the room temperature storage of equine faeces for up to 24 h and of NAP buffer for up to 150 h prior to processing.

## 1. Introduction

The current goals of the equine microbiome research field include widescale sampling efforts in order to gain a universal understanding of the microbes that inhabit the gut of the horse, with a focus on characterizing the functional properties of those organisms [1]. Faecal sampling has facilitated the investigation of the equine microbiota in the distal compartments of the gut across the last two decades and has been utilized in the exploration of laminitis [2,3,4], colitis [5], colic [6], age [7,8], obesity [7,9], weight-loss [10,11], and behavioural reactivity [12]. The study of the microbiome has revealed key insights into disease states and susceptibility across species, with research in the human field spearheading our understanding of the gut microbiome as a complex entity with an influence on immunity, metabolism, and inflammation [13]. However, studies of the faecal microbiota vary in their methodological approaches, from sample collection methodology and storage to 16S rRNA gene primer selection and sequence-clustering approaches. The influence of sampling and study design on microbial community estimations is widely acknowledged in microbiome research [14], but currently, there is no universal “Gold Standard” approach to studying the equine faecal microbiota, and individual institutions likely follow their own internally developed protocols.

One of the challenges in researching the faecal microbiome of animals can be the remote sampling locations that require faeces to be stored over a variable period of time prior to processing or analysis. In relation to the storage of equine faecal samples, studies have demonstrated that storage for as little as 6 h at ambient temperature can induce changes in the diversity and composition of the microbial community [15,16]. Studies such as these highlight the importance of the close recording of metadata and indicate that the preservation of equine faeces is required where immediate freezing is not available to ensure reliable results.

As was achieved with the Human Microbiome Project [17], the multistate equine-focused consortia (the Equine Microbiome Project) are currently working towards characterising the equine gut microbiome with statistically significant numbers [1]. Figures derived from human studies recommend a sample size of 500 per group to detect a 5% difference with 80% power [18]. For such efforts, involving the owners or carers of horses in the collection and storage of faecal samples has the potential to significantly increase sample sizes. However, doing so would benefit from accepted standard operating procedures for the handling of samples—the first milestone cited by the equine consortia. 

It is likely that, during transit, and particularly where samples require shipment across large distances, samples may be exposed to fluctuations in temperature. A potential methodology that could be employed to preserve samples under such conditions is an insulated envelope posted with the sampling receptacle along with a cool pack for storage in the owner’s freezer, which would then be packaged with the sample in the envelope for return. Alternatively, previously tested commercial sampling devices, such as Whatman FTA cards [19], could be used, although the expense associated would require a reliably superior result to be justifiable in a large study. A cost-effective “homemade” nucleic acid preservation (NAP) buffer, first described by Camacho-Sanchez et al. [20], has demonstrated efficacy in preserving stool from other species [21,22] so may offer a suitable option for studies of the horse. Finally, a safe and widely used bactericidal agent such as chlorhexidine digluconate (CLX), known for its effective antimicrobial properties [23], could be applied. This biguanide affects the cell wall and demonstrates bacteriostatic properties at low concentrations [24]. Such a solution could also be cost-effective but has not previously been tested for its use in microbiome research.

These preservation treatments represent methodologies with the potential for application in studies involving owner participation, helping to facilitate a “Citizen Science” approach to equine faecal microbiome profiling in the future. The aim of this study was to compare the efficacy of four different preservation treatments in capturing and maintaining the microbial community present in equine faeces over time under relatively stable environmental conditions (room temperature storage) as the first step towards identifying suitable methods for the preservation of equine faecal samples. The underlying hypothesis associated with this aim was that the conservation of the equine faecal microbial community structure at room temperature is dependent on the preservation method and temporal windows. The first objective was to assess the impact of the preservation treatment on the yield of DNA extracted, sequencing depth, diversity, and community composition. The second objective was to assess changes in these measures of the microbial community at intervals during storage at room temperature under the different preservation treatments. The third objective was to assess the impact of individual variability on the results.

## 2. Materials and Methods

This study received full ethical approval from the SRUC animal ethics committee (EQU AE 24-2019). A total of 10 individual animals were included (*n* = 10).

### 2.1. Sampling Methodology

A total of 15 faecal samples were collected from *n* = 10 horses in Aberdeenshire, Scotland, UK. The horses and samples comprised three groups. All animals were considered by owners and managers to be healthy, and none were receiving veterinary treatment or medication at the time of sampling. The first group (G1) was a heterogenous group of five horses, sampled on 10 February 2021, which were housed on the same premises but in different fields and on different management routines with varied supplementary feeding and preserved forage access. The second group (G2), sampled on 15 February 2021, comprised five Shetland ponies homogenous in relation to sex, age, breed, management, and feeding. The third group of samples (G3) comprised a repeated sampling, with faeces collected on five consecutive days from a single horse from group 1 (8 March 2021–12 March 2021). Table 1 provides a summary of the individual signalment and management details of the animals in each of the three groups. 

A single sample of fresh faeces was collected from each horse within 1 min of voiding. Sample was taken only from the top portion of the faeces that had not been in contact with the ground, ensuring both the outer component and centre of faecal balls were represented. Samples were collected into a sealable plastic bag before being homogenized thoroughly via manual manipulation through the bag and distributed into 4 × 20 mL universal tubes for COLD storage, 4 × 20 mL universal tubes containing 10 mL of NAP buffer (0.744 g% *w*/*v* EDTA disodium salt dihydrate, 0.735% *w*/*v* sodium citrate trisodium salt dehydrate, and 70% *w*/*v* ammonium sulphate; pH 5.2 with H_2_SO_4_ or HC [20]), 4 × 20 mL universal tubes containing 10 mL 2% CLX (Sigma-Aldrich, MO, USA, C9394), and 1 × clear 20 mL sterile universal tube intended for immediate processing as the −80 °C reference sample (-80REF). Tubes were loosely filled, ensuring the faeces stored in tubes containing solution or buffer were completely submerged. Using a sterile plastic tongue depressor, a portion of the remaining homogenized faeces was then smeared onto the centre of 4 × FTA cards, which were then folded and placed into a resealable plastic bag. The COLD samples were placed into individual insulated envelopes pre-packaged with cool packs that had been stored at −20 °C.

All samples were then transported to the laboratory, where the -80REF samples were homogenized with PBS-glycerol (2.4 g NaCl, 0.06 g KCl, 0.43 g Na_2_HPO_4_.2H_2_O, 0.072 g KH_2_PO_4_ in 100 mL water, adjusted to pH 7.4 and autoclaved) prior to being aliquoted into extraction-ready tubes and stored at −80 °C. One sample from each individual from each of the treatments was then stored at −80 °C, forming the first timepoint (TP1), representing the time that the samples had spent stored at room temperature (<1 h). One sample from each individual from each treatment was subsequently stored at −80 °C after 24 h (TP2), 72 h (TP3), and 150 h (TP4) of being stored on the bench top of the laboratory.

### 2.2. Sample Preparation

Samples were removed from −80 °C and thawed under refrigeration for 0.5–1.5 h depending on the treatment prior to DNA extraction. Samples stored in NAP buffer required additional processing to remove the viscous buffer. After thawing, NAP samples were homogenized by mixing them with a metal spatula, which was sterilized with ethanol and flame between samples. From the homogenized NAP samples, 1 g was transferred to a 2 mL microcentrifuge tube, and 1 mL of cold phosphate-buffered saline was added. The sample was again homogenized via inversion before centrifugation at 6000× *g* for 15 min [21]. The resulting supernatant was discarded, and this was repeated when the buffer remained. After this point, all faecal samples (other than the FTA cards) were processed in the same way following the extraction protocol below, wherein 0.25 g of each thawed sample was transferred into a sterile 2ml screwcap tube containing 0.1 g of 0.5 mm and 0.3 g of 0.1 mm sterile zirconia beads. 

### 2.3. DNA Extraction and 16S rRNA Gene Sequencing

The total genomic DNA (gDNA) was extracted from all faecal samples using the method first described by Yu and Morrison (2004) [25], which included multiple bead-beating steps and column-based elution. Minor adjustments were applied to optimize the procedure for equine faeces (see Appendix A). For FTA card extraction, twelve 4 mm punches were taken from the centre of the cards before extracting DNA using the QIAAMP DNA investigator kit protocol following the manufacturer’s instructions [26]. All samples were stored at −80 °C prior to extraction, none for more than 1 month. Extractions were performed in multiple batches, with a maximum of 24 samples in any given batch and a negative control included in each.

Amplicon sequence libraries targeting the microbial 16S rRNA region, as well as the fungal ITS region, were prepared using the QIAseq 16S/ITS 384-index kit (Catalogue No. 333827, QIAGen Hilden, Germany). This approach incorporated three pools of phased primer mixes targeting the V1–V2, V2–V3, V3–V4, V4–V5, V5–V7, V7–V9, and ITS1 regions of the genes. Each region was targeted with 11 forwards and 11 reverse primer sequences with 0–11 additional bases at the 5′ end in order to increase base diversity and improve amplicon quality. Libraries from a total of 266 gDNA samples were prepared following the manufacturer’s protocols. Negative batch and no template PCR controls were included, with 6 taken through to sequencing. A kit-positive control (to assess fungal ITS amplification) and a mock microbial-community-positive control (ATCC MSA-1003) were also included in triplicate. Amplicon libraries were sequenced using the MiSeq Sequencing platform (Illumina, San Diego, CA, USA) using V3 sequencing conditions for 2 × 300 bp paired-end reads, aiming for a sequencing depth of 70 k–140 k reads per sample (10–20 k per single region—6 × 16S and 1 × ITS region). Library preparation and DNA sequencing were performed at the Centre for Genome-Enabled Biology and Medicine.

### 2.4. Bioinformatics

Assessment and quality control of the reads were performed with FastQC version 0.11.8 [27], TrimGalore! Version 0.6.4 [28] with the -q 30 option, and MultiQC version 1.7 [29]. Sequence denoising, paired-end read joining, and chimaera checking and removal were performed using the DADA2 pipeline version 1.14.0 [30]. Taxonomies were assigned to corresponding amplicon sequence variants (ASVs) using the Silva reference database for bacterial sequences [31]. Results for each of the six V-regions were merged at this point. For the ITS region, the QIAGEN CLC Workbench tool version 22.0.2 (QIAGEN, Aarhus, Denmark) was used to demultiplex the sequencing data, perform the quality control, compute the amplicon sequence variants, and assign taxonomies using UNITE ITS database version 7.2 for fungal species [32].

### 2.5. Statistical Methods

To assess the impact of storage conditions over time upon the yield of DNA extracted from the samples, generalized least squares regression (GLS) was applied using the nlme package in R [33], as previously described by Menke et al. (2017). The effect of the COLD, CLX, NAP, and FTA treatments and timepoint upon DNA yield (log + 1) was tested with yield as the dependent variable and treatment, timepoint, and their interactions as independent variables. The −80REF samples were set as the intercept. Treatment was set as the variance term to account for heteroskedasticity between treatments. The impact of treatment, timepoint, and DNA yield upon sequencing depth was assessed via GLS regression to model the read counts (log-transformed and scaled) as a function of treatment, timepoint, and yield (log + 1) while controlling for individual variation with the random term (1|individual), again with the -80REF samples set as the intercept.

The relative abundance of taxa at each phylogenetic level was calculated and averaged across replicate samples from each treatment group. Alpha diversity was assessed with Shannon diversity, observed diversity, and Faith’s phylogenetic diversity (Faith’s PD) indices. Indices were scaled for comparable results across samples with different sequencing depths. Generalised additive mixed-effect models were built for scaled diversity metrics (observed diversity, Faith’s PD, and Shannon’s diversity), with each modelled as a function of treatment, timepoint, and group fixed effects. The smooth terms included were the scaled total sample reads with k = 5 degrees of freedom to model the potential non-linear effects of sequencing depth on the diversity metrics [34] and (1|individual) as a random effect smooth term to account for potential correlations between repeated measures and to allow for individual-specific variations in diversity. The Akaike Information Criterion, with correction for a small sample size (AIC) score, was used for model selection as a measure of the goodness of fit and complexity of the models. Models with the lowest AIC values were accepted as those most representative of the data structure. Odds ratios were calculated for each parameter estimate to assess the divergence of the significant terms from the intercept.

Beta diversity was assessed across all groups to evaluate the effect of treatment and timepoint on samples with and without the effect of interindividual variation. Weighted and unweighted Unifrac distances were calculated for these two sets of data, and the results were visualized by plotting principal coordinate analysis (PCoA) coordinates. Permutational analysis of variance (PERMANOVA) was applied to test for significant differences in beta diversity with additive models using the ADONIS function from the VEGAN package [35].

To identify statistically significant, differentially abundant taxa at the phylum and family levels, analysis of microbiome composition with bias correction was applied using the ANCOMBC2 package [36]. Data filtering was applied to include only taxa with a sum of occurrences > 3 and presence in >2% of the total samples. This threshold was selected to reduce the computational effort required for ANCOMBC2 processing and to reduce the false discovery rate (FDR) associated with multiple comparisons. The threshold for removal of rare taxa with a sum of occurrences < 3 was selected as a conservative filter that was not expected to significantly alter the results based on previous work that assessed the filtering of reads with <10 occurrences [37]. Filtering of taxa that occurred in fewer than 2% (5/248) of samples was applied to enhance the interpretability of findings. Identification of rare taxa was not an aim of the present study, and the experimental design was not optimised for the detection of rare taxa. First, within groups, the microbial community composition in samples collected at the first timepoint under different treatments was compared with the −80 °C reference sample. Within each group, the effect of time at room temperature under the different treatments was then assessed, with the treatment–timepoint interaction included as a fixed effect and the −80 °C reference sample set as the reference group. Tests including the treatment and timepoint were performed specifying Dunnet’s test-type modification in the ANCOMBC formula for multiple group comparisons. For each analysis, Holm’s method was applied to correct the *p*-value for family-wise errors associated with multiple comparisons. The threshold of significance for all hypothesis-based tests was set at the 5% alpha level (*p* < 0.05).

## 3. Results

### 3.1. Read Depth Was Most Influenced by DNA Yield and Individual Variation

A total of 74,312 ASVs were assigned across all samples, with a minimum of 12,145; a maximum of 47,584; and an average of 25,628 (±5665) reads per sample. The negative control samples yielded 11 reads, which were not assigned to any taxonomy and were thus removed. One sample failed sequencing and was excluded from the dataset (the COLD-treated sample from the second day of sampling of the repeated individual, stored at TP3). There were six FTA card samples that yielded <1 ng/µL DNA, and these were also excluded from the analysis. There were 14 distinct phyla identified across all samples, and 214 ASVs represented taxa resolved to the genus level. The fungal ITS1 region was successfully amplified in four positive control samples, all with high read depth (20,000–34,119 merged reads per control sample). In contrast, the experimental samples had either no or a very low number of ITS1 reads, except for one sample with 2,045 reads. Because of failure at the amplification stage, no further data regarding the fungal ITS region will be reported herein.

The yield of DNA extracted from the FTA cards was not directly comparable to the other methods involving direct extraction from 0.25 g of faecal material (Figure 1A). For the FTA cards, DNA was directly extracted from <0.01 g of FTA card discs from which the weight attributable to raw material was negligible, and as such, yield (ng/g) was low in this treatment. Otherwise, yields from the remaining treatments were comparable (Figure 1B). Generalised linear mixed-effect models of scaled read counts controlling for individual variation revealed the significant effect of yield on the read depth. The optimal model (AIC = 842.23) included only yield as a fixed effect, which had a coefficient of 0.220 (SE = 0.056, t = 3.909, *p* < 0.01), indicating a significant positive effect on read depth for a one-unit increase in yield scaled. Inclusion of the timepoint factor did not improve model fit (AIC = 844.38); however, estimates for read depth at timepoint 3 across treatments were significantly lower than at timepoint 1 (coefficient = −0.381, SE = 0.154, t = −2.462, *p* = 0.014), as is clearly demonstrated by the raw read plots (Figure 1C) and model estimates (Figure 1D).

### 3.2. Alpha Diversity Was Significantly Lower in FTA Cards

For each of the scaled Shannon’s, Faith’s phylogenetic, and observed diversity metrics, the best model fit comprised group and group–treatment interaction as significant fixed effects when controlling for scaled read depth and individual variation (Appendix A).

As demonstrated in Figure 2A, group 2 positively deviated from the intercept (group 3), while group 1 negatively deviated, indicating the higher and lower diversity observed in these two groups, respectively, when compared with group 3 samples. Specifically, Faith’s PD (t = −2.725, *p* = 0.007), observed (t = −2.836, *p* = 0.005), and Shannon’s (t = −2.271, *p* = 0.024) diversity were all significantly lower in group 1.

As is reflected in the raw diversity metrics (Figure 1C), regardless of the estimate, the FTA cards had consistently lower diversity compared with the -80REF samples, while the COLD, CLX, and NAP diversity measures remained relatively consistent with the -80REF. As shown in Figure 2B, all treatments tended to negatively deviate from the -80REF sample; however, model estimates indicated that this was not significant in any treatment other than FTA cards (see Appendix A Appendix A for parameter estimates). It should be noted that the -80REF was sampled only at the first timepoint, while the treatment estimates included samples from across the time trial. Faith’s PD (t = −2.474, *p* = 0.014), observed (t = −2.631, *p* = 0.009), and Shannon’s (t = −2.881, *p* = 0.004) diversity were all significantly lower in the FTA cards. However, compared with group 3, the FTA treatment within group 2 was significantly higher for all metrics: Faith’s PD (t = 3.150, *p* = 0.002), observed (t = 3.218, *p* = 0.001), and Shannon’s (t = 2.292, *p* = 0.023) diversity.

### 3.3. Beta Diversity Was Most Influenced by Individual and Group Membership

Across the 10 unique individuals comprising groups 1 and 2, individual and group were the only significant terms in an additive model assessing the influence of treatment (*p* = 0.118), timepoint (*p* = 0.634), group (*p* = 0.001), and individuals (*p* = 0.001) on weighted Unifrac distances. While significant, this was not clearly represented in the PCoA plot (Figure 3A). In the unweighted Unifrac model, treatment was significant (*p* = 0.001), in addition to group and individuals. The separation of individuals was distinguishable in the unweighted Unifrac PCoA plots, as was the separation of treatment for the FTA samples from group 2 specifically (Figure 3B). It should be noted that the first two axes of the PCoA represented only a small, combined proportion of variance (7%).

Within group 3, which was used to test for the effects of treatment and timepoint in the absence of individual variability, beta diversity measured using weighted Unifrac distance was not significantly influenced by either treatment or timepoint (Figure 3C). Unweighted Unifrac models identified treatment (*p* = 0.001) and timepoint (*p* = 0.002) but not their interaction term (*p* = 0.112) as significant drivers of beta diversity. Again, this effect was not clearly represented by PCoA (Figure 3D), with no visual clustering of samples based on treatment or timepoint.

### 3.4. Microbial Community Was Distinct in FTA Cards

After data filtering, the taxa count across samples ranged between 196 and 3723 unique reads, with an average of 1458.714 (±573.641) reads per sample. The relative abundance of taxa overall and across treatments is summarised in Appendix A. The effect of treatment at the outset was tested by comparing the samples collected at the first timepoint (0 h room temperature) with -80REF samples. Treatment significantly impacted the taxa captured in equine faecal samples, and across the three groups, the FTA cards demonstrated an altered community profile compared with other treatments (Figure 4).

In groups 1 and 2, the FTA cards were found to have a lower abundance of taxa from the phylum Fibrobacterota, as well as from the family *Eubacteriaceae* from the Firmicutes phylum. In group 2, taxa from the phyla Bacteroidetes and Spirochaetes were also significantly lower in the FTA card samples, while Actinobacteria were significantly higher. In this group, the taxa of the families *Spirochaetaceae*, *p-251-o5*, *Fibrobacteraceae*, and *Coriobacteriales Incertae Sedis* were also significantly lower in the FTA cards than in the -80REF samples. In common with group 2, the FTA card samples in group 3 also had a lower abundance of Spirochaetes and *Spirochaetaceae*. However, in this group, differences in NAP buffer and CLX treatment were also highlighted. Compared with the reference, the phylum Fibrobacterota was higher in NAP, and Patescibacteria was higher in CLX samples. The results are shown in Table 2.

### 3.5. NAP Inhibited Time-Associated Changes in Bloom Taxa Abundance 

The raw abundance of the dominant phyla across all groups over time can be viewed in Appendix A. The effect of time was tested by comparing treatments across the timepoints tested with the -80REF samples. Comparisons were again made within each group (Table 3). The group found to have comparatively higher alpha diversity (group 2) had the highest number of differentially abundant taxa, and the patterns of growth and decline over time of some of these taxa were also noted in groups 1 and 3.

Within group 1, Proteobacteria was significantly higher in the CLX treatment at TP3 compared with the -80REF samples. At the family level, *Clostridiaceae* was significantly higher in the TP2 and TP3 COLD samples, and *Planococcaceae* was higher at the third timepoint in both COLD and CLX samples. The family *Moraxellaceae* was also significantly higher in TP3 under the CLX treatment.

Group 2 also showed significant increases in Proteobacteria in CLX at TP3, and a higher abundance of this phylum was also identified in COLD-treated samples at TP3 and TP4. In addition, Bacteroidetes were significantly decreased in the FTA card samples at TP2, and Fibrobacterota had declined in COLD samples by TP4. At the second timepoint, the FTA cards had significantly fewer Spirochaetes. Family-level changes were also observed at the third and fourth timepoints for the COLD and CLX samples. As was found for group 1, the group 2 COLD samples had higher *Planococcaceae* and *Morraxellaceae* at TP3 and, in addition, showed decreased *Fibrobacteraceae* by TP4. By the third timepoint, *Clostridiaceae* was again significantly higher in CLX samples in group 2.

In group 3, the Actinobacteria phylum was significantly more abundant than the reference samples in the FTA cards at the second timepoint and in both the FTA and COLD samples by the third. The *[Eubacterium] coprostanoligenes* family was also more abundant in the FTA samples by the second timepoint in this group. As for groups 1 and 2, *Clostridiaceae* was found to have significantly increased by timepoints 3 and 4 in both the CLX and COLD treatments in group 3.

## 4. Discussion

The present study demonstrated the significant effect of selected storage conditions on the community structure of the equine faecal microbiota. The results demonstrated that the storage of equine faecal samples in an insulated envelope with a cold pack in NAP or CLX solution effectively preserved the microbial community for up to 24 h at room temperature. Collection in NAP buffer was associated with a higher abundance of taxa from the phylum Fibrobatcerota and in CLX solution with a higher abundance of Patescibacteria, which may indicate the improved capture of these taxa using these treatments, even compared with the rapidly stored -80 reference sample. Only NAP buffer inhibited the significant changes in microbial abundance that occurred by 72 h of room temperature storage. The results also showed that the FTA cards may not have effectively captured the microbial community present in equine faecal samples; however, the ineffective extraction of DNA from the FTA cards may have impacted these results. The majority of the difference in taxa observed between samples was attributable to individuals more than the effect of storage-based factors. These findings present cold storage, NAP buffer, and 2% CLX solution as acceptable treatments for the preservation of equine faecal samples ensured to be stored at −80 °C or processed for DNA extraction within 24 h of sampling, with NAP buffer being the preferred treatment where longer delays until processing (up to 150 h) could occur.

Alpha diversity in equine faecal samples remained relatively stable over time in relation to the preservation method applied, similar to the results found in canine faecal samples under preservation [38]. Previous work investigating equine faecal-sampling methodologies without preservation has demonstrated that time until processing is a major driver of microbial diversity in samples [15,16], and as such, the combined results of these studies support the use of some form of preservation for equine faecal samples where immediate storage or snap freezing is not available. In contrast to studies using untreated samples that found a linear increase in alpha diversity over time, the preserved samples stored over time had lower alpha diversity than the -80 reference samples. Despite this overall effect, timepoint (e.g., time at room temperature) was not significant in any of the tested models.

Time at room temperature was only a significant driver of beta diversity in the unweighted Unifrac model applied to test the effect of treatment and time on the individual repeated samples (group 3). These results suggest that the effect of time at room temperature on the microbial community composition was more pronounced in terms of the presence/absence of taxa (which was captured in unweighted Unifrac) rather than relative abundance (which was captured in weighted Unifrac). Further, the significance of the timepoint term in beta diversity models but not alpha diversity ones suggests that temporal variation in the community composition was not directly linked to changes in the diversity within individual samples (alpha diversity). Regardless of the effect of preservation, the predominant factors associated with both alpha and beta diversity were the group and individual factors, which is not uncommon in methodological studies of the faecal microbiota [39,40,41,42].

Samples from the homogenous group (group 2) of animals kept under identical and consistent management (in a university herd setting), as well as the repeated samples taken from an individual (group 3), had a higher relative alpha diversity than samples from the “heterogenous” livery yard sample group (group 1). This effect may be similar to that seen in studies wherein higher microbial diversity is observed in the faeces of semi-feral equine populations compared with conventionally managed ones [43,44]. In the present study, the population with the higher faecal microbial diversity was not semi-feral; however, it was not exposed to a conventional livery yard environment, ridden exercise, frequent transport, or stabling, whereas the livery yard group was, which may account for some of this effect. It was not the intention of this study to evaluate environmental factors associated with faecal microbial diversity, and as such, the sample size limits further investigation of these effects in the present data. It is worthwhile to note that, despite demonstrating low alpha diversity across all metrics in groups 1 and 3, the FTA cards captured significantly higher alpha diversity within group 2—the higher diversity group. This could suggest that, despite performing poorly in comparison with the other treatments, the FTA cards were still sensitive enough to capture higher diversity in a group comparison scenario. 

Commercial treatments such as RNAlater and Ogene.GUT have demonstrated efficacy in preserving microbial communities in stool samples from species other than equines [45]. However, for widescale applications, such methods may become costly as the sample size increases. Homemade NAP buffer [20] has proved effective in the preservation of sheep [21] and human faeces [22] and may offer an affordable alternative preservation method for studies in the equine field. Here, NAP buffer effectively suppressed the increase in abundance of the suspected bloom of taxa from the families *Clostridiaceae*, *Moraxellaceae*, and *Planococcaceae*, which occurred in the COLD and CLX samples. The latter two of these taxa were observed in a previous study to have increased in equine faecal samples by 24 h at room temperature, whereas the former, *Clostridiaceae,* was found to increase after 96 h in that study. In addition, NAP buffer appeared to favour the preservation of *Ruminococcaceae* and prevent its reduction by 72 h at room temperature, which occurred under other treatments. This loss of *Ruminococcaceae* was previously reported to occur within a 12 h delay of equine faecal sample processing [15,16]. 

The CLX treatment did not appear to exert a preservative effect on the equine faecal microbiota at room temperature storage beyond 24 h. However, it did yield a higher abundance of taxa belonging to the phylum Patescibacteria within group 3. Chlorhexidine’s activity is pH-dependent [24], and it may have lost efficacy with time as the samples underwent fermentation caused by increasingly abundant facultative anaerobes and aerobes [15]. Testing its efficacy at higher concentrations may be warranted owing to the fact that the CLX treatment preserved the microbiota over 24 h and demonstrated no significant deviance from the reference samples within this time.

An apparently discriminatory profile was captured by FTA cards over the 150-h storage trial. Richness and evenness, as measured using Shannon’s, Faith’s phylogenetic, and observed diversity, were lower in the FTA cards compared with the other treatments. This is in contrast with studies on other species that found FTA cards captured the highest diversity [40]. While FTA cards were effective in preventing some of the bloom taxa consistent with those previously identified [15], the under-representation of the Fibrobacterota and Spirochaetes phyla, lower alpha diversity, and the distinct profile from other sample treatments warrant further investigation. The reasons for these results are most likely due to low DNA yields resulting from ineffective DNA extraction. Given the highly fibrous nature of some equine faecal samples, it is also possible that the microbes present in the faeces were not adequately exposed to the FTA card material. Methods for improved DNA yield and microbial community capture from FTA cards have since been tested and optimized [46]. Replicating the present findings and testing DNA extraction methodologies would greatly improve our understanding of the utility of FTA cards for equine faecal sampling.

The failure of ITS region DNA amplification during the library preparation stage highlighted the need to optimize DNA extraction methods for equine faecal samples, particularly where studies intend to include fungal gene surveys. The method used to extract DNA from equine faeces was first outlined by Yu and Morrison [25] for the extraction of DNA from rumen samples, and it was later recommended as the standard protocol for DNA extraction from human stool samples [47]. In a study comparing multiple methods of DNA extraction [48], this method was recommended as being suitable for studies of the fungal microbiome owing to the extensive bead-beating steps included. It is also comparable to methods used in studies of the mycology of bovine rumen [49] and equine faeces [50]; however, the DNA extraction protocol used in these studies included a 90 °C incubation between bead-beating steps, as opposed to the present study’s 70 °C incubation. It may be the case that this temperature and the precise conditions used herein were not adequate enough to lyse fungal cell walls to liberate fungal DNA. Future studies may consider including steps to optimize procedures for fungal DNA extraction under individual laboratory settings.

While NAP buffer appears to be effective, it should be noted that the DNA extraction methodology had to be adapted to account for the density of the buffer (see Section 2). The addition of saline and subsequent centrifugation is necessary to remove buffer from the sample and is a step also applied with RNAlater^®^ preparations. As the authors of a previous work using NAP buffer noted [21], there may be a loss of cellular material at this step. However, the total read count was not impacted by the NAP treatment, nor were there any taxa present at significantly lower abundances in the NAP-buffer-treated samples compared with the controls. Thus, the effect of pre-extraction processing on these samples does not appear to have affected the data. Regardless, it is crucial to optimize DNA extraction protocols for equine faeces in NAP buffer to account for the density of the buffer and ensure reliable results. Phenol–chloroform-based extraction, which has been assessed for use in equine faeces, may be a preferable approach for this step [51]. Exploration of the effect of CLX at higher concentrations upon the microbial community is also warranted owing to the ease of use of this solution. The time taken for the cold pack to reach room temperature was not assessed; however, a previous study using a “cool-box” storage scenario noted significant fluctuations in temperature up to 32 h, at which point, the temperature met that of the room [41]. External temperatures and the cool pack selected are likely to influence the effectiveness of this treatment; thus, the further development of methods to keep samples cool during storage may improve the reliability of this approach. 

It is also important to note that the study conditions tested herein are not representative of the extensive fluctuations in environmental temperatures, including freeze–thaw cycles and extreme heat, which may be encountered during the shipment of samples. Studies testing such extreme conditions have shown the effective preservation of human and dog faeces using FTA cards and preservatives, including RNAlater^®^ [19]. Future studies can build upon the findings of the present work by assessing the effectiveness of the preservation methods explored in this study in horse faeces under more extreme environmental conditions, emulating those that may be encountered during transportation.

## 5. Conclusions

The results presented here suggest that homemade nucleic acid preservation (NAP) buffer may be a suitable agent for the prolonged storage (up to 150 h) of equine faeces collected under field conditions. Storage with a cool pack, in NAP buffer, or in CLX solution effectively preserved the microbial community for up to 24 h. In contrast, FTA cards resulted in a distinct microbial profile, with a significant under-representation of taxa from the Fibrobacterota phylum. This study highlights the importance of preservation methods for accurate faecal microbial analysis and suggests the need for the further optimization of DNA extraction methods for FTA cards, as well as fungal gene surveys in equine faeces. Based on these findings, it is recommended that posting equine faecal samples stored in NAP buffer, 2% CLX solution, or in an insulated envelope with a cold pack could be utilized for microbiome studies assessing diversity and dominant taxa prevalence where 24 h delivery is ensured and the environmental temperature is equivalent to room temperature. Where >24 h storage is predicted, NAP buffer may prevent the overgrowth of *Clostridiaceae*, *Planococcaceae,* and *Moraxellaceae* and the loss of *Ruminococcaceae*. To firmly establish the most effective equine faecal preservation method for field transport, it is imperative that these findings are replicated and validated under real-world conditions that simulate the transportation process.

## Figures and Tables

**Figure 1 animals-13-03107-f001:**
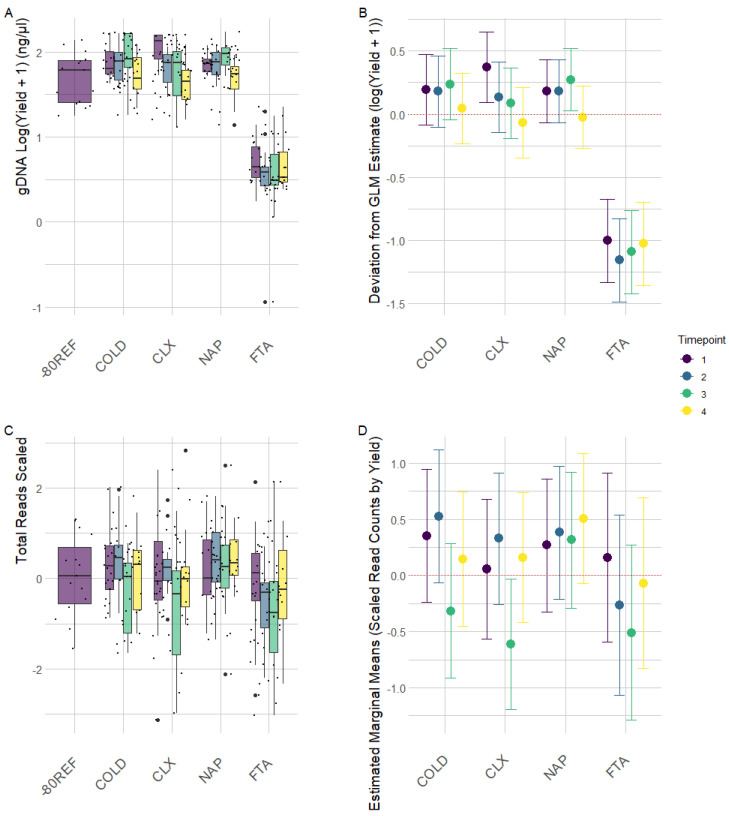
Raw data and generalized linear mixed-effect model estimates for DNA yield (**A**,**B**) and read depth (**C**,**D**) from equine faecal samples stored with a cool pack (COLD) with chlorhexidine digluconate solution (CLX), nucleic acid preservation buffer (NAP), and FTA cards (FTA) after being stored for 0 (TP1), 24 (TP2), 72 (TP3), and 150 (TP4) hours at room temperature. (**A**) Log-transformed (+1) yield of DNA extracted from samples based upon preservation treatment across time. Black points represent transformed data points. (**B**) Parameter estimates from a generalized linear model of DNA yield (log + 1) as a function of sample preservation treatment controlling for the treatment × time at room temperature interaction, with bars showing the estimates’ 95% confidence intervals. (**C**) Scaled read counts by treatment across time. Black points represent scaled data points. (**D**) Generalized linear model estimates of scaled read counts as a function of sample treatment and time at room temperature while controlling for individual variation with bars showing the estimates’ 95% confidence intervals.

**Figure 2 animals-13-03107-f002:**
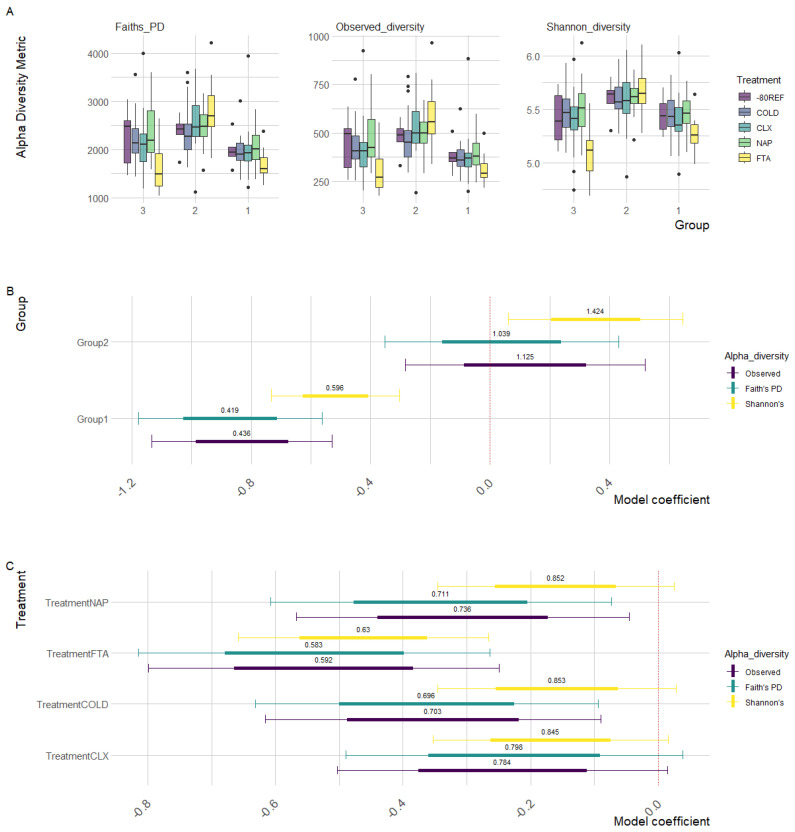
Generalised additive mixed-effect model parameters of scaled diversity metrics (observed diversity, Faith’s PD, and Shannon’s diversity) modelled as a function of the treatment controlling the effects of individual horses and the scaled total sample reads. Points represent parameter estimates with 95% confidence intervals denoted by the bordering thin lines and standard error by thick lines, with the intercept set as the 0 h-80REF reference sample. Odds ratios calculated for each parameter estimate represent the magnitude of divergence of the corresponding treatment from −80 reference samples. (**A**) Raw diversity estimates between treatments (boxes) within groups. (**B**) Parameter estimates for the group. (**B**,**C**) Parameter estimates for the treatment.

**Figure 3 animals-13-03107-f003:**
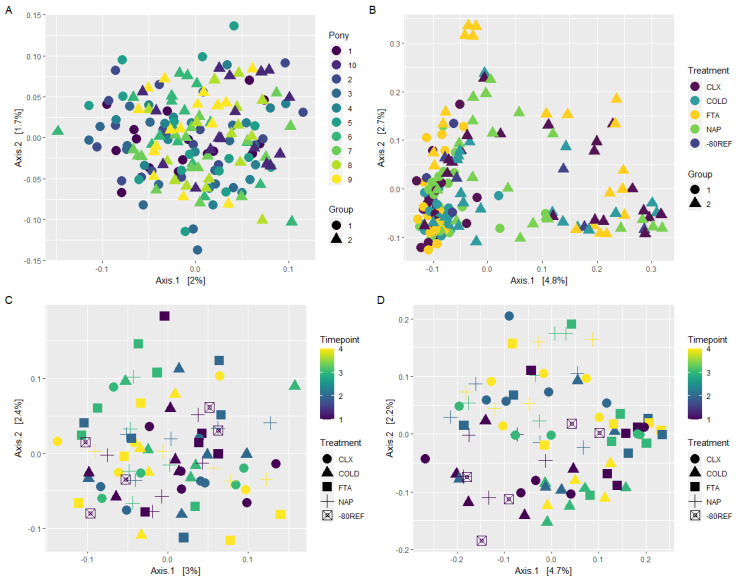
Principal coordinate analysis (PCoA) of weighted (**A**,**C**) and unweighted (**B**,**D**) Unifrac distances. (**A**,**B**) Samples from the 10 unique individuals in groups 1 and 2. (**C**,**D**) Within individual sampling from one individual across 5 days (group 3).

**Figure 4 animals-13-03107-f004:**
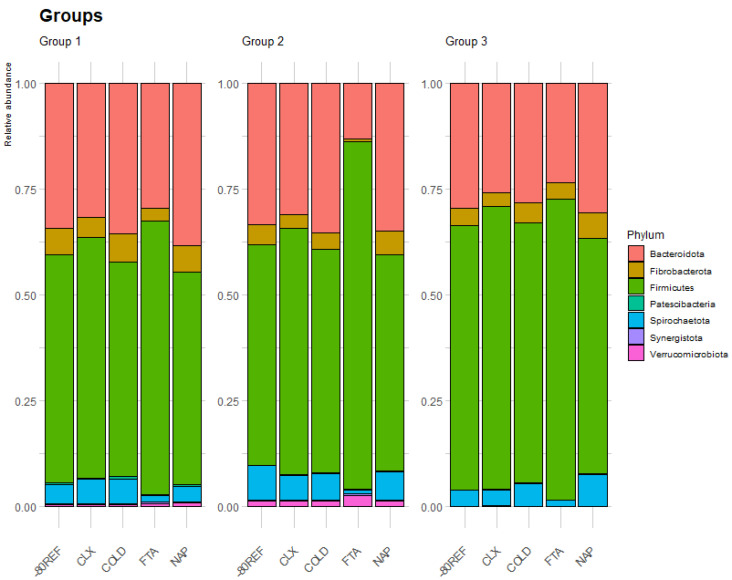
Relative abundance of the eight most abundant phyla averaged by treatment within the first timepoint (0 h stored at room temperature prior to −80 °C storage) across the three groups.

**Table 1 animals-13-03107-t001:** Signalment and management of horses included in the study.

Horse	Group	Sex	Breed	Age (Years)	Routine
1	1	Mare	Irish draft X	9	Grazing 24/7, stabled for exercise. Fed Dengie Alfa A, Allen & Page Calm and Condition, TopSpec Cool Balancer, NAF Seaweed and Garlic Granules (once per day). Haylage when stabled. Hay in field.
2	1	Gelding	WB X	8	Grazing day, stabled at night w/haylage. Fed Spillers Conditioning Fibre and TopSpec UlsaKind Cubes (twice per day).
3	1	Gelding	Irish draft X	15	Grazing day, stabled at night w/ haylage. Fed TopSpec Comprehensive Feed Balancer, Speedibeet, NAF 5 Star Supaflex Joint Supplement (twice per day).
4	1	Mare	Irish Cob	17	Grazing 24/7 w/hay, stabled for exercise, fed Dengie HiFi Molasses Free and Allen & Page Veteran Vitality. Haylage when stabled (once per day).
5	1	Gelding	Haflinger	11	Grazing during day, stabled at night w/hay. FedTopspec Lite balancer and Topspec TopChop Zero and Linseed oil (twice per day).
6	2	Gelding	Shetland	14	Restricted grazing 24/7 with rationed hay access. Fed Baileys Fibre Plus Nuggets.
7	2	Gelding	Shetland	14	Restricted grazing 24/7 with rationed hay access. Fed Baileys Fibre Plus Nuggets.
8	2	Gelding	Shetland	11	Restricted grazing 24/7 with rationed hay access. Fed Baileys Fibre Plus Nuggets.
9	2	Gelding	Shetland	13	Restricted grazing 24/7 with rationed hay access. Fed Baileys Fibre Plus Nuggets.
10	2	Gelding	Shetland	13	Restricted grazing 24/7 with rationed hay access. Fed Baileys Fibre Plus Nuggets.
2	3 (D1)	Gelding WB X 8 Grazing day, stabled at night w/haylage. Fed Spillers Conditioning Fibre and TopSpec UlsaKind Cubes (twice per day).			
	3 (D2)				
	3 (D3)				
	3 (D4)				
	3 (D5)				

**Table 2 animals-13-03107-t002:** Results of ANCOMBC2 showing taxa with significant differences between treatments prior to room temperature storage.

	Taxa	Log Fold Change	SE	Wald’s Test Statistic	*p*-Value	q-Value	Differential to -80REF
Group 1							
Phylum	*Fibrobacterota*	−1.09	0.367	−2.965	0.003	0.036	FTA
Family	*Eubacteriaceae*	3.421	0.768	4.455	<0.001	<0.001	FTA
Group 2							
Phylum	*Bacteroidetes*	−1.014	0.318	−3.191	0.001	0.011	FTA
	*Spirochaetes*	−2.368	0.373	−6.351	<0.001	<0.001	FTA
	*Fibrobacterota*	−2.415	0.470	−5.139	<0.001	<0.001	FTA
	*Actinobacteria*	5.133	1.078	4.764	<0.001	<0.001	FTA
Family	*Spirochaetaceae*	−2.368	0.433	−5.464	<0.001	<0.001	FTA
	*p-251-o5*	−3.134	0.717	−4.372	<0.001	<0.001	FTA
	*Fibrobacteraceae*	−2.415	0.519	−4.651	<0.001	<0.001	FTA
	*Coriobacteriales Incertae Sedis*	5.450	0.734	7.428	<0.001	<0.001	FTA
	*Eubacteriaceae*	5.909	0.577	10.238	<0.001	<0.001	FTA
Group 3							
Phylum	*Fibrobacterota*	0.641	0.194	3.308	0.001	0.010	NAP
	*Spirochaetes*	−1.118	0.308	−3.635	<0.001	0.003	FTA
	*Patescibacteria*	3.339	1.126	2.967	0.003	0.033	CLX
Family	*Spirochaetaceae*	−1.1181	0.336	−3.327	0.001	0.040	FTA

**Table 3 animals-13-03107-t003:** Results of ANCOMBC2 using Dunnet’s test-type modification for multiple group comparisons between the four storage treatments at four timepoints and the reference samples.

	Taxa	Log Fold Change	SE	Wald’s Test Statistic	*p*-Value	q-Value	Differential to -80REF
Group 1							
Phylum	*Proteobacteria*	5.771	0.957	6.033	<0.001	<0.001	TP3CLX
Family	*Planococcaceae*	4.589	1.041	4.409	<0.001	0.001	TP3CLX
		4.372	1.041	4.201	<0.001	0.003	TP3COLD
	*Moraxellaceae*	5.701	1.024	5.565	<0.001	<0.001	TP3CLX
	*Clostridiaceae*	4.143	1.017	4.074	<0.001	0.005	TP3COLD
		4.529	1.017	4.455	<0.001	0.001	TP3COLD
Group 2							
Phylum	*Fibrobacterota*	−5.683	1.25	−4.546	<0.001	<0.001	TP4COLD
	*Bacteroidetes*	−3.109	1.144	−2.717	0.007	0.028	TP2FTA
	*Proteobacteria*	2.745	0.762	3.603	<0.001	0.008	TP3CLX
		5.148	0.762	6.758	<0.001	<0.001	TP3COLD
		5.193	0.762	6.818	<0.001	<0.001	TP4COLD
	*Spirochaetes*	−4.305	1.338	−3.218	0.001	0.036	TP1FTA
Family	*Planococcaceae*	5.182	0.887	5.839	<0.001	<0.001	TP3COLD
	*Fibrobacteraceae*	−5.683	1.332	−4.266	<0.001	<0.001	TP4COLD
	*Moraxellaceae*	4.799	0.862	4.807	<0.001	<0.001	TP3COLD
		4.142	0.862	4.807	<0.001	<0.001	TP4COLD
	*Clostridiaceae*	3.102	0.845	3.671	<0.001	0.006	TP3CLX
Group 3							
Phylum	*Actinobacteria*	3.315	0.781	4.246	<0.001	<0.001	TP2FTA
		2.291	0.781	2.934	0.003	0.023	TP3COLD
		2.952	0.781	3.782	<0.001	0.001	TP3FTA
Family	*Clostridiaceae*	2.919	0.963	3.032	0.002	0.046	TP3CLX
		3.238	0.963	3.364	0.001	0.015	TP4CLX
		3.099	0.963	3.219	0.001	0.024	TP3COLD
		5.91	1.021	3.032	<0.001	<0.001	TP4COLD
	*[Eubacterium] coprostanoligenes group*	4.211	1.38	3.052	0.002	0.043	TP2FTA

## Data Availability

All data files are available from SRUC’s FigShare repository: https://doi.org/10.58073/SRUC.24126477.v1 (accessed on 14 September 2023).

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
