# Peer review of "Homemade Nucleic Acid Preservation Buffer Proves Effective in Preserving the Equine Faecal Microbiota over Time at Ambient Temperatures"

_animals, 2023, doi:10.3390/ani13193107_

Round 1

Reviewer 1 Report

In general, the study entitled “ Homemade nucleic acid preservation buffer proves effective in preserving the equine faecal microbiota over time at ambient temperatures” is really interesting, well described and discussed. My minor comments aim to increase the scientific soundness and clarity of it.

 My comments:

Line 29 – In Nomina Anatomica Veterinaria the term “distal hindgut” or “hindgut” in not present. Please use the correct name.

Line 87 – replace with “first described by Camacho-Sanchez et al. [19], “

Line 96 – please present your hypothesis.

Line 107 – please provide any details concerning general veterinary status of the experimental animals.

Line 137 – “-80 °C reference samples” were already abbreviated to “-80REF”  (line 131)

Line 399 – NAP and CLX acronyms were used for the fisrt time in line 86 and line 90 (respectively)

Author Response

In general, the study entitled “ Homemade nucleic acid preservation buffer proves effective in preserving the equine faecal microbiota over time at ambient temperatures” is really interesting, well described and discussed. My minor comments aim to increase the scientific soundness and clarity of it.

We thank you for your prompt review of this manuscript, and for your constructive feedback. We agree that the suggested edits improve the clarity of the manuscript. Below are the revisions applied in order to address your comments (line numbers are in reference to the revised version of the document).

 My comments:

Line 29 – In Nomina Anatomica Veterinaria the term “distal hindgut” or “hindgut” in not present. Please use the correct name.

  • Corrected use of “hind gut” (line 31, line 52, line 54-55) changing to “gut” or “distal compartments of the gut”

Line 87 – replace with “first described by Camacho-Sanchez et al. [19], “

  • Line 91: Replaced with first described by Camacho-Sanchez et al. [19]

Line 96 – please present your hypothesis.

  • Line 104 – 107: Added hypothesis

Line 107 – please provide any details concerning general veterinary status of the experimental animals.

  • Line 118 – 120: Added details of veterinary status of horses

Line 137 – “-80 °C reference samples” were already abbreviated to “-80REF”  (line 131)

  • Line 117: Abbreviated to -80REF

Line 399 – NAP and CLX acronyms were used for the fisrt time in line 86 and line 90 (respectively)

  • Line 422-433: Amended to retain only abbreviated treatments rather than repeating

Reviewer 2 Report

In the manuscript the Authors describe their research concerning the best nucleic acid stabilizing agent for field studies, while testing the impact of the agents in preserving the microbiota composition in horse feces. The research is fluently written, the design well explained and altogether gives a clear message of the best option in biological material preservation. I have only a few comments and questions.

1. In the Introduction, the Authors concentrate on horse feces analysis presentation, while more global approach could be also explained here. Some information about other samples (human, canine..) is presented in Discussion section, however, I think a more broad scope could be presented in the Introduction section as well. That would immediately increase the impact of the data presented.

2. The samples were kept for a certain time at room temperature, and then put to -80. Why was the DNA not extracted immediately after incubation at room temperature, thus eliminating the effects of freezing altogether? Also, the methods describe, that the samples were thawed under refrigeration, but it is unclear if it was closely monitored, or timed while thawing.

3. Line 153 presents the method of removal of storage buffer, as the washing the sample with saline, and removing the supernatant. Could the DNA from the lysed cells be also removed in this step and could it influence the results?

4. The DNA extraction protocol for FTA cards was different from other samples. Could it be that the protocol also influenced the final results, not only the sample stabilizing capability of the FTA cards?

 5. Why the threshold of including a taxa in the analysis was set to occurrences > 3 and presence in > 2% of the total samples?

6. The figures presented could be "cleaner" - e.g. y axis labels should clearly represent the value displayed.

7. Line 36 "16S rRNA" should be capital S.

8. ASV abbreviation is not explained in the text.

Author Response

In the manuscript the Authors describe their research concerning the best nucleic acid stabilizing agent for field studies, while testing the impact of the agents in preserving the microbiota composition in horse feces. The research is fluently written, the design well explained and altogether gives a clear message of the best option in biological material preservation. I have only a few comments and questions.

We thank you for your prompt and careful consideration of this manuscript. We agree that highlighting work across species in the introduction improves the impact of the work, and are grateful that you were able to highlight aspects of the methodology that required further clarification. Namely, justification for filtering the data prior to ancom analysis, the effect of saline washing of NAP samples, and the impact of the apparent failure of DNA extraction from FTA cards on the results. We have added description and a more considered justification for these aspects of the work (as below). In addition, we thank you for highlighting the need to improve figure labels, and have done so for Figures 1 and 2, where clarity was required. We have found the incorporation of your suggestions to improve the quality and rigor of the manuscript.

In the Introduction, the Authors concentrate on horse feces analysis presentation, while more global approach could be also explained here. Some information about other samples (human, canine..) is presented in Discussion section, however, I think a more broad scope could be presented in the Introduction section as well. That would immediately increase the impact of the data presented.

    • Line 57 – 60: Added a sentence and reference around the human microbiome: 

      Ref: Hou, K., Wu, ZX., Chen, XY. et al. Microbiota in health and diseases. Sig Transduct Target Ther 7, 135 (2022). https://doi.org/10.1038/s41392-022-00974-4

2. The samples were kept for a certain time at room temperature, and then put to -80. Why was the DNA not extracted immediately after incubation at room temperature, thus eliminating the effects of freezing altogether? Also, the methods describe, that the samples were thawed under refrigeration, but it is unclear if it was closely monitored, or timed while thawing.

  • Thank you for raising this good point. We felt that in the interest of minimising inter-day variation with regard to DNA extractions, and to replicate the approach taken upon returning from field sampling (which may be to store the sample in the freezer immediately upon receipt in order to batch samples together for extraction and processing), that storage at -80 prior to extraction was acceptable where all samples underwent this procedure. However we acknowledge that the optimal approach would be immediate extraction. 
  • Line 158: Added description of sampling thawing- this was not explicitly timed or monitored, rather samples were place in the fridge in the morning and were extracted once things were set up. This is a point that I hadn’t considered monitoring, but will in future work.

3. Line 153 presents the method of removal of storage buffer, as the washing the sample with saline, and removing the supernatant. Could the DNA from the lysed cells be also removed in this step and could it influence the results?

  • Lines 531 – 537: Added description of effect of saline wash on cell and DNA in the discussion section. An important consideration- and absolutely may have had an impact on results. The analysis presented did not detect stark differences between NAP and "COLD" samples at the first timepoint- so we suspect that, at least within analytical error of our analysis, this was not influencial. 

4. The DNA extraction protocol for FTA cards was different from other samples. Could it be that the protocol also influenced the final results, not only the sample stabilizing capability of the FTA cards?

  • Line 24-25: Added sentence in simple summary and discussion (Line 430-431) to highlight ineffective processing of FTA cards. Agreed, this protocol is likely responsible for a proportion of the differences observed. 

 5. Why the threshold of including a taxa in the analysis was set to occurrences > 3 and presence in > 2% of the total samples?

  • Lines 243 – 252: Added description of justification to filter data at defined threshold in the methods section. Largely to do with computing power, and lack of a robust design to make statements about any rare taxa with confidence using ANCOM analysis. 
  • Lines 358-360: Added a description of the data resulting from data filtering for full transparency in results 3.4. Microbial community was distinct in FTA cards

6. The figures presented could be "cleaner" - e.g. y axis labels should clearly represent the value displayed.

  • Line 290: Figure 1a – 1d – y-axis labels changed
  • Line 325: Figure 2 altered for clarity and y axis labels changed

7. Line 36 "16S rRNA" should be capital S.

  • Capitalised 

8. ASV abbreviation is not explained in the text.

  • Line 200: Added the full description of ASVs

Reviewer 3 Report

The authors compare different preservation solutions for equine samples. The work is generally well performed, but the authors did not test real field conditions that would make the results applicable. As you state in the final conclusion, your results are applicable when "environmental temperature is  equivalent to room temperature", but this will be almost be never the case in a real field situation. Temperatures vary between summer and winter and regions in the world. Temperatures in a delivery truck in the summer will always be higher than room temperature (much higher). In winter you may experience freeze thaw cycles when samples are shipped. Shipping can often be delayed by one day. So your conclusions are only valid if shipping is perfectly controlled. This needs to be more prominently described in the conclusions of abstract, also in introduction and discussion.

Furthermore, please add a supplemental table that lists the actual relative abundances for all samples all all phylogentic levels. Fold changes only deliver partial information and make it difficult to compare results across studies 

Author Response

The authors compare different preservation solutions for equine samples. The work is generally well performed, but the authors did not test real field conditions that would make the results applicable. As you state in the final conclusion, your results are applicable when "environmental temperature is  equivalent to room temperature", but this will be almost be never the case in a real field situation. Temperatures vary between summer and winter and regions in the world. Temperatures in a delivery truck in the summer will always be higher than room temperature (much higher). In winter you may experience freeze thaw cycles when samples are shipped. Shipping can often be delayed by one day. So your conclusions are only valid if shipping is perfectly controlled. This needs to be more prominently described in the conclusions of abstract, also in introduction and discussion.

Furthermore, please add a supplemental table that lists the actual relative abundances for all samples all all phylogentic levels. Fold changes only deliver partial information and make it difficult to compare results across studies 

Thank you for your prompt review of this manuscript, and for highlighting a key aspect of this work which we agree was requires to be made clear to avoid over-selling these results as being currently applicable in across all “real-world” scenarios. We have made adjustments to highlight further the fact that this work is representative of sample storage at room temperature, and that we did not test shipping conditions. Respectfully, while we absolutely agree upon the need to highlight the limitations in applying the results outside of the room temperature context – especially in conclusions-  we feel that within the limited 250 word abstract, the room temperature aspect of the work is highlighted sufficiently at present, so have not altered the abstract in this regard.

We have added a table of the relative abundance data to the supplementary materials as requested. We are grateful for your observations, and feel that emphasis of the limitations of the work, and provision of supplementary data, as suggested, has improved the manuscript overall.

  • Line 27: Highlighted that the study demonstrates room temperature findings simple description
  • Line 83-85: Added statements in paragraphs 4 & 5 of introduction to highlight that fluctuations in temperature which samples under shipment may be exposed to.
  • Lines 102-104: highlighted that this study represents a step towards identifying suitable treatments for sample storage, and highlights that the study aimed to test stable temperatures.
  • Line 538-539: Relative abundance data in supplementary signposted: in results 4. Microbial community was distinct in FTA cards
  • Lines 423-424: 1st paragraph of discussion, added room temperature
  • Lines 548-555: Added paragraph 10 to the discussion, added reference to study which tested at fluctuating temperatures akin to shipping. Highlighted need for further validation to identify preferred treatment for field study where samples may be exposed to extreme conditions.
  • Lines 571-573: Added sentence in conclusion to highlight that further validation needed.
  • Added two supplementary tables with Supplementary Table 8 showing relative abundance for all samples, (up to) top 30 across taxonomic ranks from phylum to genus. Also added supplementary table 9 which shows relative abundance grouped by treatment.

Round 2

Reviewer 3 Report

Thanks for addressing my comments